# WHAT IS IMAGE CAPTIONING MADE OF?

## ABSTRACT

We hypothesize that end-to-end neural image captioning systems work seemingly well because they exploit and learn 'distributional similarity' in a multimodal feature space, by mapping a test image to similar training images in this space and generating a caption from the same space. To validate our hypothesis, we focus on the 'image' side of image captioning, and vary the input image representation but keep the RNN text generation model of a CNN-RNN constant. We propose a sparse bag-of-objects vector as an interpretable representation to investigate our distributional similarity hypothesis. We found that image captioning models (i) are capable of separating structure from noisy input representations; (ii) experience virtually no significant performance loss when a high dimensional representation is compressed to a lower dimensional space; (iii) cluster images with similar visual and linguistic information together; (iv) are heavily reliant on test sets with a similar distribution as the training set; (v) repeatedly generate the same captions by matching images and 'retrieving' a caption in the joint visual-textual space. Our experiments all point to one fact: that our distributional similarity hypothesis holds. We conclude that, regardless of the image representation, image captioning systems seem to match images and generate captions in a learned joint image-text semantic subspace.

## 1 INTRODUCTION

Image description generation, or image captioning (IC), is the task of automatically generating a textual description for a given image. The generated text is expected to describe, in a single sentence, what is visually depicted in the image, for example the entities/objects present in the image, their attributes, the actions/activities performed, entity/object interactions (including quantification), the location/scene, etc. (e.g. "*a man riding a bike on the street*").

Significant progress has been made with *end-to-end* approaches to tackling this problem, where large-scale parallel image–description datasets such as Flickr30k (Young et al., 2014) and MSCOCO (Chen et al., 2015b) are used to train a CNN-RNN based neural network IC system (Vinyals et al., 2016; Karpathy & Fei-Fei, 2015; Xu et al., 2015). Such systems have demonstrated impressive performance in the COCO captioning challenge[1] according to automatic metrics, seemingly even surpassing human performance in many instances (e.g. CIDEr score > 1.0 vs. human's 0.85) (Chen et al., 2015a). However, in reality, the performance of end-to-end systems is still far from satisfactory according to metrics based on human judgement[2]. Thus, despite the progress, this task is currently far from being a solved problem.

In this paper, we challenge the common assumption that end-to-end IC systems are able to achieve strong performance because they have learned to 'understand' and infer semantic information from visual representations, i.e. they can for example deduce that "*a boy is playing football*" purely by learning directly from mid-level image features and the corresponding textual descriptions in an implicit manner, without explicitly modeling the presence of *boy*, *ball*, *green field*, etc. in the image. It is believed that the IC system has managed to infer that the phrase *green field* is associated with some 'green-like' area in the image and is thus generated in the output description, or that the word *boy* is generated because of some CNN activations corresponding to a young person. However, there seems to be no concrete evidence that this is the case. Instead, we hypothesize that the apparently

---

[1] http://cocodataset.org/#captions-challenge2015
[2] http://cocodataset.org/#captions-leaderboard

strong performance of end-to-end systems is attributed to the fact that they are exploiting the *distributional similarity* in the multimodal feature space. To our best knowledge, our paper gives the first empirical analysis on visual representations for the task of image captioning.

What we mean by 'distributional similarity' is that IC systems essentially attempt to match images from the training set that is most similar to a test image, and generate a caption from the most similar training instances (or generate a 'novel' description from a combination of training instances, for example by 'averaging' the descriptions). Previous work has alluded to this observation (Karpathy, 2016; Vinyals et al., 2016), but it has not been thoroughly investigated. This phenomena could also be in part attributed to the fact that the datasets are repetitive and simplistic, with an almost constant and predictable linguistic structure (Lebret et al., 2015; Devlin et al., 2015; Vinyals et al., 2016).

In this paper we investigate the hypothesis of *distributional similarity* in IC by focusing on the *image* side of image captioning. Most previous work has concentrated on the *text* side of image captioning, e.g. by optimizing the language modelling capabilities of the RNN (Rennie et al., 2016; Liu et al., 2017) to improve its performance on automatic metrics. While there have been efforts on improving IC by utilizing or modeling images more effectively, for example by using attention over mid-level image features (Xu et al., 2015) and high-level object proposals (Anderson et al., 2017), in this work we are specifically interested in interpretability and we focus on using a simpler (and faster) model for empirical evaluation. We explore the basic yet effective CNN-RNN model (Karpathy & Fei-Fei, 2015), and investigate the representational contributions while keeping the RNN generator constant. More advanced models can be considered specific variants of Karpathy & Fei-Fei (2015).

It is worth noting that we are interested in demonstrating the phenomenon of distributional similarity in IC, rather than achieving or improving state-of-the-art performance, As such, we do not resort to fine-tuning or extensive hyperparameter optimization or ensembles. Therefore, our model is not comparable to state-of-the-art models such as Vinyals et al. (2016), which optimize IC by fine-tuning the image representations, exploring beam size, scheduled sampling, and using ensemble models. Instead, we vary only the image representation to demonstrate that end-to-end IC systems utilize distributional similarity on the image side to generate captions, regardless of the image representation used.

Our main contributions are:

- **An IC experiment** where we vary the input image representation but keep the RNN text generation model constant (Section 3). This experiment demonstrates that regardless of the image representation (a continuous image embedding or a sparse, low-dimensional vector), end-to-end IC systems seem to utilize a visual-semantic subspace for IC.
- The introduction of a simple, sparse **bag-of-objects representation** that contains information about the presence of objects in the images. We use this as a tool to investigate the contribution of images in the image captioning framework.
- The introduction of **pseudo-random vectors** derived from object-level representations as a means to evaluate IC systems. Our results show that end-to-end models in this framework are remarkably capable of separating structure from noisy input representations.
- An experiment where IC models are conditioned on image representations **factorized and compresssed to a lower dimensional space** (Section 4.1). We show that high dimensional image embeddings that are factorized to a lower dimensional representation and used as input to an IC model result in virtually no significant loss in performance, further strengthening our claim that IC models perform similarity matching rather than image understanding.
- An **analysis of different image representations and their transformed representations** (Sections 4.2 and 4.3). We visualize the initial visual subspace and the learned joint visual semantic subspace and observe that the visual semantic subspace has learned to cluster images with similar visual and linguistic information together, further validating our claims of distributional similarity.
- An experiment where the IC model is **tested on an out-of-domain dataset** (Section 4.4), which has a slightly different image distribution. We observe that models, including the state-of-the-art models, show a better performance on test sets that have a similar distribution as the training. However, their performance deteriorates when the distributions are slightly different.
- An analysis on the **uniqueness of captions** generated by IC models using different image representations (Section 4.5). We hypothesize that the captions are often repeated as

they are usually generated by matching images in the joint space and retrieving a relevant caption. Our experiments validate this claim.

Overall, the study suggests that regardless of the representation used, end-to-end IC models implicitly learn and exploit multimodal similarity spaces rather than performing actual image understanding.

This study is in line with the recent work that explore understanding of deep learning models and the representational interpretations (Papernot et al., 2016; Szegedy et al., 2013; Sturm, 2014) and works that have tried to delve into the image captioning task (Devlin et al., 2015; Vinyals et al., 2016). To the best of our knowledge, ours is the first work that investigates IC focusing specifically on image representations and their effects.

## 2 MODEL SETTING

For the experiments in Section 3, we base our implementation on a simple end-to-end approach by Karpathy & Fei-Fei (2015). We use the LSTM (Hochreiter & Schmidhuber, 1997) based language model as described in Zaremba et al. (2014).

To condition the image information, we first perform a linear projection of the image representation followed by a non-linearity:

$$Im_{feat} = \sigma(W \cdot I_m)$$

Here, $I_m \in \mathcal{R}^d$ is the $d$-dimensional initial image representation, $W \in \mathcal{R}^{n \times d}$ is the linear transformation matrix, $\sigma$ is the non-linearity. We use Exponential Linear Units (Clevert et al., 2015) as the non-linear activation in all our experiments. Following Vinyals et al. (2015), we initialize the LSTM based caption generator with the projected image feature.

**Training and Inference**    The caption generator is trained to generate sentences conditioned on the image representation. We train the model by minimizing the cross-entropy, i.e., the sentence-level loss corresponds to the sum of the negative log likelihood of the correct word at each time step:

$$\Pr(S|Im_{feat};\theta) = \sum_t \log(\Pr(w_t|w_{t-1}..w_0; Im_{feat})) \tag{1}$$

where $\Pr(S|Im_{feat};\theta)$ is the sentence-level loss conditioned on the image feature $Im_{feat}$ and $\Pr(w_t)$ is the probability of the word at time step $t$. This is trained with standard teacher forcing as described in Sutskever et al. (2014) where the correct word information is fed to the next state in the LSTM.

Inference is typically performed with approximation techniques like beam search or sampling (Karpathy & Fei-Fei, 2015; Vinyals et al., 2015). In this paper, as we are mainly interested in the studying effect of different image representations, we focus on the language output that the models can most confidently produce. Therefore, in order to isolate any other variables from the experiments, we generate captions using a greedy argmax based approach for consistency (unless stated otherwise, we always use greedy decoding).

## 3 IMAGE CAPTIONING WITH DIFFERENT IMAGE REPRESENTATIONS

In this section, we verify our hypothesis that a 'distributional similarity' space exist in end-to-end IC systems. Such systems attempt to match image representations in order to condition the RNN decoder to generate captions that are similar to the closest images, rather than actually understanding the image in order to describe the image. We keep the IC model constant (Section 2) across experiments, and vary only the image representation used.

### 3.1 IMAGE REPRESENTATIONS

#### 3.1.1 LOWER-BOUND REPRESENTATION

**Random:** We condition the LSTM with a 300-dimensional vector comprising random values sampled uniformly between $[0, 1)^3$. This feature essentially gives us a worst case image feature and provides an artificial lower bound.

#### 3.1.2 REPRESENTATIONS FROM IMAGE-LEVEL CLASSIFICATION

The following pre-trained CNNs are used:

- *VGG19* (Simonyan & Zisserman, 2015) pre-trained on ILSVRC (Russakovsky et al., 2015).
- *ResNet152* (He et al., 2016) also pre-trained on ILSVRC.
- *Places365-ResNet152* (Zhou et al., 2014), a variant of *ResNet152* pre-trained on the Places2 dataset (Zhou et al., 2017). We investigate whether scene-specific categories are useful for IC without the network being trained to classify object-specific categories.
- *Hybrid1365-ResNet152* (Zhou et al., 2014), a *ResNet152* variant trained on the concatenation of the ILSVRC and Places2 datasets and predicts both object and scene classes.

We explore various representations derived from the CNNs above:

**Penultimate layer (*Penultimate*):** Most previous attempts for IC use the output of the penultimate layer of a CNN pre-trained on ILSVRC. Previous work motivates using 'off-the-shelf' feature extractors in the framework of transfer learning (Razavian et al., 2014; Donahue et al., 2014). Such features have often been applied to image captioning (Mao et al., 2014; Karpathy & Fei-Fei, 2015; Xu et al., 2015; Gao et al., 2015; Vinyals et al., 2015; Donahue et al., 2015) and have been shown to produce state-of-the-art results. Therefore, for each image, we extract the ***fc7*** layer of *VGG19* ($4096D$) and the ***pool5*** layer for the *ResNet152* variants ($2048D$) .

**Class prediction vector (*Softmax*):** We also investigate higher-level image representations, where each element in the vector is an estimated posterior probability of object categories. Note that the categories may not directly correspond to the captions in the dataset. While there are alternative methods that fine-tune the image network on a new set of object classes extracted in ways that are directly relevant to the captions (Fang et al., 2015; Wu et al., 2016), we study the impact of off-the-shelf prediction vectors on the IC task. The intuition is that category predictions from pre-trained CNN classifiers may also be beneficial for IC, alongside the standard approach of using mid-level features from the penultimate layer. Therefore, for each image, we use the predicted category posterior distributions of *VGG19* and *ResNet152* for 1000 object categories), *Places365-ResNet152* (365 scene categories), and *Hybrid-ResNet152* (1365 object and scene categories).

**Object class word embeddings (*Top-$k$*):** Here we experiment with a method that utilizes the averaged word representations of top-$k$ predicted object classes. We first obtain *Softmax* predictions using *ResNet152* for 1000 object categories (synsets) per image. We then select the objects that have a posterior probability score $> 5\%$ and use the 300-dimensional pre-trained word2vec (Mikolov et al., 2013) representations[4] to obtain the averaged vector over all retained object categories. This is motivated by the central observation that averaged word embeddings can represent semantic-level properties and are useful for classification tasks (Arora et al., 2016).

#### 3.1.3 REPRESENTATIONS FROM OBJECT-LEVEL DETECTIONS

We also explore representing images using information from object *detectors* that identifies *instances* of object categories present in an image, rather than a global, image-level classification. This can potentially provide for a richer and more informative image representation. For this we use:

- *ground truth* (***Gold***) region annotations for instances of 80 pre-defined categories provided with MSCOCO. It is worth noting that these were annotated independently of the image

---

[3] We also tried using 1,000-dimensions, yielding similar results, albeit slightly poorer.
[4] https://code.google.com/archive/p/word2vec/

captions, i.e. people writing the captions had no knowledge of the 80 categories and the annotations. As such, there is no direct correspondence between the region annotations and image captions.

- a state-to-the-art object detector *YOLO* (Redmon & Farhadi, 2016), pre-trained on MSCOCO for 80 categories (***YOLO-Coco***), and on MSCOCO and ILSVRC for over 9000 categories (***YOLO-9k***).

We explore several representations derived from instance-level object class annotations/detectors above:

**Bag of objects (*BOO*):**   We represent each image as a sparse 'bag of objects' vector, where each element represents the frequency of occurrence for each object category in the image (***Counts***). We also explore an alternative representation where we only encode the presence or absence of the object category regardless of its frequency (***Binary***), to determine whether it is important to encode object counts in the image. These representations help us examine the importance of explicit object categories and in a sense interactions between object categories (*dog* and *ball*) in the image representation. We investigate whether such a sparse and high-level *BOO* representation is helpful for IC. It is also worth noting that *BOO* is different from the *Softmax* representation above as it encodes the *number* of object occurrences, not the *confidence* of class predictions at image level. We compare *BOO* representations derived from the ***Gold*** annotations (***Gold-Binary*** and ***Gold-Counts***) and both ***YOLO-Coco*** and ***YOLO-9k*** detectors (***Counts*** only).

**Pseudo-random vectors:**   To further probe the capacity of the model to discern image representations in an image distributional similarity space, we propose a novel experiment where we examine a type of representation where *similar images are represented using similar random vectors*, which we term as *pseudo-random vectors*. We form this representation from ***BOO Gold-Counts*** and ***BOO Gold-Binary***. Formally, $Im_{feat} = \sum_{o \in \text{Objects}} f \times \phi_o$, where $\phi_o \in \mathcal{R}^d$ is an object-specific random vector and $f$ is a scalar representing counts of the object category. In the case of ***Pseudorandom-Counts***, $f$ is the frequency counts from ***Gold-Counts***. In the case of ***Pseudorandom-Binary***, $f$ is either 0 or 1 based on ***Gold-Binary***. We use $d = 120$ for these experiments.

## 3.2   Datasets and experimental setup

**Dataset**   We evaluate image captioning conditioned on different representations on the most widely used dataset for IC, *MSCOCO* (Chen et al., 2015b). The dataset consists of $82,783$ images for training, with at least five captions per image, totaling to $413,915$ captions. We perform model selection on a 5000-image development set and report the results on a 5000-image test set using standard, publicly available splits[5] of the MSCOCO validation dataset as in previous work (Karpathy & Fei-Fei, 2015).

**Evaluation Metrics**   We evaluated system outputs using the standard evaluation metrics for image captioning using the most common metrics: BLEU (Papineni et al., 2002) which is computed from 1-gram to 4-gram precision scores (B-1 $\cdots$ B-4), Meteor (Denkowski & Lavie, 2014) (M) and CIDEr (Vedantam et al., 2015) (C) and SPICE (Anderson et al., 2016) (S). All these metrics are based on some form of $n$-gram overlap between the system output and the reference captions (i.e. no image information is used). For each system-generated caption, we compare against five references. We used the publicly available *cocoeval* script for evaluation.[6]

**Model Settings and Hyperparameters**   We use a single hidden layer LSTM with 128-dimensional word embeddings and 256-dimensional hidden dimensions. As training vocabulary we retain only words that appear at least twice.

## 3.3   Image Captioning Results

We report results of IC on MSCOCO in Table 1, where the IC model (Section 2) is conditioned on the various image representations described in Section 3.1. As expected, using random image

---

[5]http://cs.stanford.edu/people/karpathy/deepimagesent
[6]https://github.com/pdollar/coco

| | Representation | B-1 | B-2 | B-3 | B-4 | M | C | S |
|---|---|---|---|---|---|---|---|---|
| | Random | 0.48 | 0.24 | 0.11 | 0.07 | 0.11 | 0.07 | 0.03 |
| Softmax | VGG19 | 0.62 | 0.43 | 0.29 | 0.19 | 0.20 | 0.61 | 0.13 |
| | ResNet152 | 0.62 | 0.43 | 0.29 | 0.19 | 0.20 | 0.62 | 0.12 |
| | Places365-ResNet152 | 0.60 | 0.41 | 0.28 | 0.19 | 0.19 | 0.56 | 0.12 |
| | Hybrid1365-ResNet152 | 0.60 | 0.41 | 0.27 | 0.18 | 0.19 | 0.60 | 0.12 |
| Penultimate | VGG19 (fc7) | 0.65 | 0.46 | 0.32 | 0.22 | 0.21 | 0.69 | 0.14 |
| | ResNet152 (Pool5) | 0.66 | 0.48 | 0.33 | 0.23 | 0.22 | 0.74 | 0.15 |
| | Places365-ResNet152 | 0.61 | 0.41 | 0.27 | 0.19 | 0.19 | 0.55 | 0.12 |
| | Hybrid1365-ResNet152 | 0.65 | 0.46 | 0.32 | 0.23 | 0.22 | 0.72 | 0.14 |
| Embeddings | Top-$k$ | 0.62 | 0.42 | 0.28 | 0.19 | 0.20 | 0.63 | 0.13 |
| BOO | Gold-Binary | 0.65 | 0.47 | 0.32 | 0.22 | 0.22 | 0.75 | 0.15 |
| | Gold-Counts | 0.67 | 0.48 | 0.33 | 0.23 | 0.22 | 0.81 | 0.16 |
| | YOLO-Coco | 0.65 | 0.46 | 0.32 | 0.22 | 0.22 | 0.75 | 0.15 |
| | YOLO-9k | 0.64 | 0.45 | 0.31 | 0.21 | 0.20 | 0.68 | 0.13 |
| Pseudo-random | Pseudorandom-Binary | 0.65 | 0.46 | 0.31 | 0.21 | 0.21 | 0.73 | 0.14 |
| | Pseudorandom-Counts | 0.67 | 0.48 | 0.34 | 0.23 | 0.22 | 0.80 | 0.15 |

Table 1: Results on the MSCOCO test split, where we vary only the image representation and keep other parameters constant. The captions are generated with $beam = 1$. We report **B**LEU (1-4), **M**eteor, **C**IDEr and **S**PICE scores.

.

embeddings clearly does not provide any useful information and performs poorly. The *Softmax* representations with similar sets of object classes (*VGG19*, *ResNet152*, and *Hybrid1365-ResNet152*) have very similar performance. However, the *Places365-ResNet* representations perform poorly. We note that the posterior distribution may not directly correspond to captions as there are many words and concepts that are not contained in the set of object classes. Our results differ from those by Wu et al. (2016) and Yao et al. (2016) where the object classes have been fine-tuned to correspond directly to the caption vocabulary. We posit that the degradation in performance is due to spurious probability distributions over object classes for similar looking images.

The performance of the *Pool5* image representations shows a similar trend for *VGG19*, *ResNet152*, and *Hybrid1365-ResNet152*. *ResNet152* is slightly better in performance. The *Places365-ResNet* representation performs poorly. We posit that the representations from the image network trained on object classes rather than scene classes are able to capture more fine-grained image details from the images, whereas the image network trained with scene-based classes captures more coarse-grained information.

The performance of the averaged top-$k$ word embeddings is similar to that of the *Softmax* representation. This is interesting, since the averaged word representational information is mostly noisy: we combine top-$k$ synset-level information into one single vector, however, it still performs competitively.

We observe that the performance of the *Bag of Objects (BOO)* sparse 80-dimensional annotation vector is better than all other image representations judging by the CIDEr score. We remark here again, that this is despite the fact that the annotations may not directly correspond to the semantic information in the image or the captions. The sparse representational information is indicative of the presence of only a subset of potentially useful objects. We notice two distinct patterns, a marked difference with *Binary* and *Count* based representations. This takes us back to the motivation that image captioning ideally requires information about objects, interaction between the objects with attribute information. Although our representation is really sparse on the object interactions, it captures the basic concept of the presence of more than one object of the same kind, and thus provides some kind of extra information. A similar trend is observed by Yin & Ordonez (2017), although in their models they further try to learn interactions using a specified object RNN.

We also notice that predicted objects using *YOLOCoco* performs better than the *YOLO9k*. This is probably expected as the *YOLOCoco* was trained on the same dataset hence obtaining better object proposals. We also observed that with *YOLO9k*, we had a significant number of objects being predicted for the test images that were not seen with the training set (around 20%).

| Method | B-1 | B-2 | B-3 | B-4 | M | C | S |
|--------|-----|-----|-----|-----|---|---|---|
| PCA | 0.66 | 0.48 | 0.34 | 0.24 | 0.22 | 0.75 | 0.15 |
| ICA | 0.66 | 0.48 | 0.34 | 0.24 | 0.22 | 0.74 | 0.15 |
| PPCA | 0.66 | 0.48 | 0.34 | 0.24 | 0.22 | 0.76 | 0.15 |
| FULL | 0.66 | 0.48 | 0.33 | 0.23 | 0.22 | 0.74 | 0.15 |

Table 2: Performance of compressed Pool5 representations

| Model | B-1 | B-2 | B-3 | B-4 | M | C |
|-------|-----|-----|-----|-----|---|---|
| Pool5 | 0.60 | 0.41 | 0.26 | 0.17 | 0.14 | 0.29 |
| SC | 0.62 | 0.42 | 0.28 | 0.18 | 0.17 | 0.35 |
| TDBU | 0.60 | 0.40 | 0.26 | 0.17 | 0.17 | 0.34 |

Table 3: Performance of models on Flickr30k

The most surprising result is the performance of the pseudo-random vectors. We notice that both the *pseudo-random binary* and the *pseudo-random count* based vectors perform almost as good as the *Gold* objects. This suggests that the conditioned RNN is able to remove noise and learn some sort of a common 'visual-linguistic' semantic subspace.

## 4 ANALYSIS ON DISTRIBUTIONAL SIMILARITY IN IC

We perform further analysis on the different image representations to gain a further understanding of the representations and demonstrate our distributional similarity hypothesis.

### 4.1 FACTORIZING THE REPRESENTATIONS

In Section 3.3, we observed encouraging results from the bag of objects based representations despite being sparse, low-dimensional, and only partially relevant to captions. Interestingly, using pseudo-random vectors derived from bag of objects also gave excellent performance despite the added noise. This leads to the question: are high-dimensional vectors necessary or relevant? To answer this, we evaluate whether the performance of the model significantly degrades if we reduce the dimensionality of a high dimensional representation. We experiment with three exploratory factor analysis based methods – Principal Component Analysis (PCA) (Halko et al., 2011), Probabilistic Principal Component Analysis (PPCA) (Tipping & Bishop, 1999) and Independent Component Analysis (ICA) (Hyvärinen et al., 2004). In all cases, we obtain 80-dimensional factorized representations on *ResNet152 pool5* ($2048D$) that is commonly used in IC. We summarize our experiment in Table 2. We observe that, the representations obtained by all the factor models seem to retain the necessary representational power to produce appropriate captions equivalent to the original representation. This seems contradictory as we expected a loss in the information content when compressing to arbitrary 80-dimensions. This experiment indicates that the model is not explicitly utilizing the full expressiveness of the full 2048-dimensional representations. We conclude that the model is able to learn from a seemingly weak, structured information and is able to result in a performance that is close to one that uses the full representation.

### 4.2 ANALYZING IMAGE REPRESENTATIONS

In this section, we investigate the distributional similarity hypothesis by inspecting the regularities in the initial representation state for several representations from Section 3.1, using the interpretable bag-of-objects representation. If the representation is informative for IC, then the representations should ideally semantically related images together, and in turn allow for relevant captions to be generated.

We compare different image representations with respect to their ability to group and distinguish between semantically related images. For this, we selected three categories from MSCOCO ("dog", "person", "toilet") and also pairwise combinations of these ("dog+person", "dog+toilet", "person+toilet"). Up to 25 images were randomly selected for each of these six groups (single category or pair) such that the images are annotated with *only* the associated categories. Each group is represented by the average image feature of these images. Figure 1 shows the cosine distances between each group, for each of our image representations. The *Bag of Objects* model clusters these groups the best, as expected (e.g. the average image representation of "dog" correlates with images containing "dog" as a pair like "dog+person" and "dog+toilet"). The *Softmax* models seem to also to exhibit semantic clusters, although to a lesser extent. This can be observed with "person", where the features are not semantically similar to any other groups. The most likely reason is that there is no "person" category in ILSVRC. Also, *Place365* and *Hybrid1365 Softmax* (Figure 1c) also showed

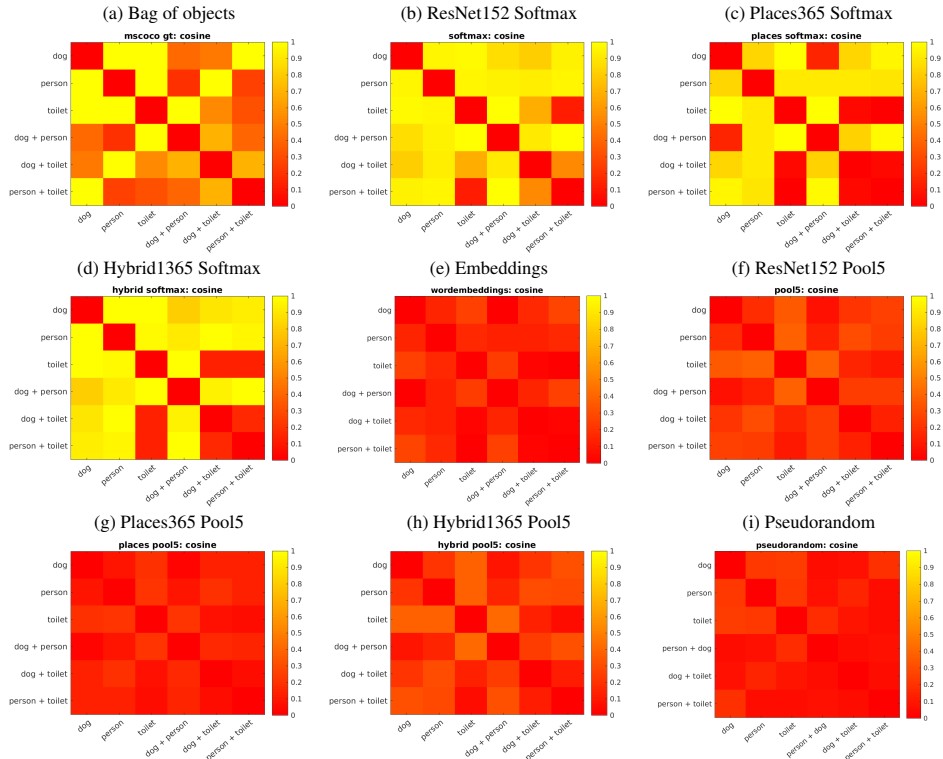

Figure 1: The cosine distance matrix between six groups (three MSCOCO categories and pairwise combinations of the three categories) from the train dataset. Each group is represented by the average image feature of 25 randomly selected images from the category or combination of categories.

very strong similarity for images containing "toilet", where or not they contain "dog" or "person", possibly because they capture scene features. On the other hand, *Pool5* features seem to result in images that are more similar to each other than *Softmax* overall.

## 4.3 ANALYZING TRANSFORMED IMAGE REPRESENTATIONS

Considering our earlier hypothesis as proposed in Section 3.3 that the conditioned RNN is learning some sort of a common 'visual-linguistic' semantic space, we explore the difference in representations in the initial representational space and the transformed representational space. The transformation is learned jointly as a subtask of the image captioning. We posit that image representations in the transformed space will be more semantically coherent with respect to both images and captions. To visualize the two representational spaces, we use *Barnes-Hut t-SNE* (Maaten & Hinton, 2008) to compute a 2-dimensional embedding over the test split.

In general, we found that images are initially clustered by visual similarity (*Pool5*) and semantic similarity (*Softmax*, *Bag of Objects*). After transformation, we observe that some linguistic information from the captions has resulted in different types of clusters.

Figure 2 highlights some interesting observations of the changes in clustering across three different representations. For *Pool5*, images seem to be clustered by their visual appearance, for example snow scenes in Figure 2a, regardless of the subjects in the images (people or dogs). After transformation, separate clusters seem to be formed for snow scenes involving a single person, groups of people, and dogs. Interestingly, images of dogs in fields and snow scenes are also drawn closer together.

*Softmax* (Figure 2b) shows many small, isolated clusters before transformation. After transformation, bigger clusters seem to be formed – suggesting that the captions have again drawn related images together despite being different in the *Softmax* space.

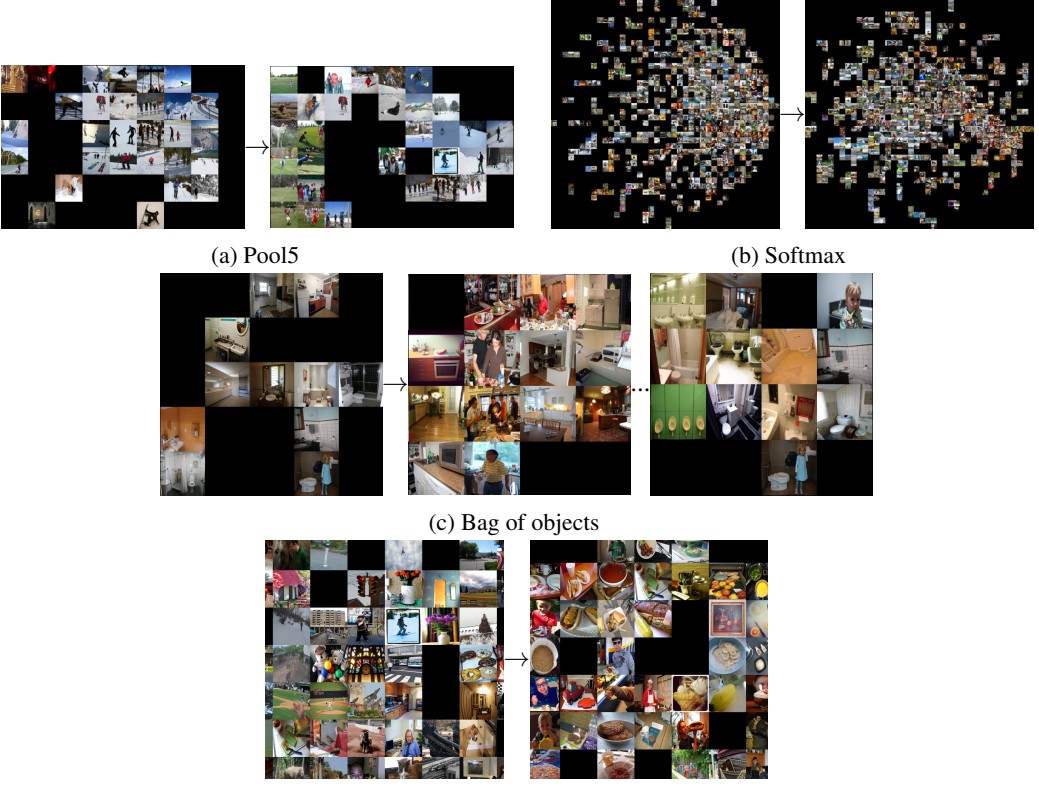

(a) Pool5          (b) Softmax

(c) Bag of objects

(d) Pseudo-random

Figure 2: Visualization of the t-SNE projection of initial representational space (left) vs. the transformed representational space (right). See main text for a more detailed discussion.

For *bag of objects* (Figure 2c), objects seem to be clustered by co-occurrence of object categories, for example toilets and kitchens are clustered since they share sinks. Toilets and kitchens seem to be further apart in the transformed space.

A similar observation was made by Vinyals et al. (2016) in which the authors observe that end-to-end based image captioning models are capable of performing retrieval tasks with comparable performance to the task specific models that are trained with the ranking loss.

We further perform a similar analysis on the pseudorandom representations (Figure 2d). We observe that the initial representations have very little explicit information and do not cluster. The projected representations however form clusters that mimic the projected space of the bag-of-object cluster.

Full sized version of images in Figure 2 are presented anonymously in: `https://github.com/anonymousiclr/HJNGGmZ0Z`

### 4.4 DOMAIN DEPENDENCY

We now demonstrate that end-to-end models are heavily reliant on datasets that have a similar training and test distribution. We posit that an IC system that performs similarity matching will not perform well on a slightly different domain for the same task. Demonstrating this will further validate our hypothesis that IC systems perform image matching to generate image captions.

Thus, we evaluate several models trained on MSCOCO on 1000 test image samples from the *Flickr30k* (Young et al., 2014) dataset [7]. Like MSCOCO, Flickr30k is an image description dataset; however, unlike MSCOCO, the images have a different distribution and the descriptions are slightly longer and more descriptive.

---

[7]the test split is obtained from http://staff.fnwi.uva.nl/d.elliott/wmt16/splits.zip

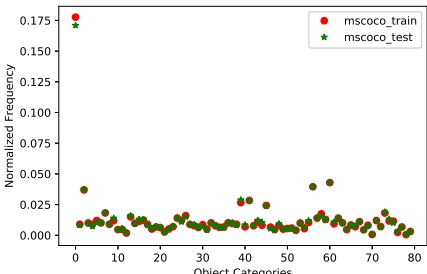 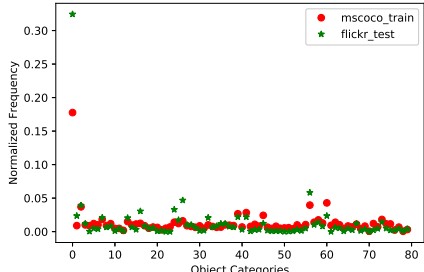

Figure 3: Distributions over train and test sets    Figure 4: Distributions over train and Flickr test

We evaluate the captions generated by our model with *Resnet152* pool5 representation and by two other state-of-the-art models pretrained on MSCOCO: a) Self-Critical (SC) (Rennie et al., 2016) based on self critical sequence training that uses reinforcement learning using metrics, and b) Bottom Up and Top Down (TDBU) (Anderson et al., 2017) based on top-down and bottom-up attention using object region proposals. Both the state-of-the-art models are much more complex than the image conditioned RNN based language model. The results are summarized in Table 3.

We observe that the scores drop by a large margin. A similar observation was made by Vinyals et al. (2016), and they alluded the drop in scores to the linguistic mismatch between the datasets. However, we probed the out of training vocabulary words in the Flickr30k test set and observed that it was around 8.6% which seems to be the usual unseen rate. This suggests that there is more to the issue than mere vocabulary mismatch. We observe that while typical sentences on Flickr30k are structurally different and are generally longer, the model is still unable to generate good bigrams or even unigrams as is evident from B-1 and B-2 scores in Table 3.

We further investigated the distributions of objects in the images by using YOLO object detector (trained on MSCOCO). We first detect objects on the MSCOCO training set and our MSCOCO test set followed by objects on the Flickr30k test set. It Figure 3 we show the normalized frequency versus the distribution of objects over the MSCOCO train and test sets. We notice that the distributions are very similar and mostly overlap. In Figure 4 we show the normalized frequency versus distribution of objects detected over MSCOCO train and Flickr30k test sets. We observe that the two distributions are slightly different, they don't overlap as closely as we see in Figure 3. We hypothesize that the difference in distribution is one of the difference that reflects in the lower performance of a model that is trained on the MSCOCO dataset performing poorly on the Flickr30k test set.

## 4.5    UNIQUENESS OF CAPTIONS

We postulate that image captions are often repeated because they are generated by 'retrieving' similar images in the joint image-text semantic space and generating the relevant caption at this point in the space.

To investigate this, we first show that, regardless of the representation, end-to-end IC systems do *not* generate unique captions for every distinct image. We use the full validation portion of the MSCOCO dataset (40,504 images) and produce captions with four types of distinct image representations. We report the results in Table 4. We observe that in almost all cases, the produced representations are far from unique. In most cases, there is a significant portion of the captions that are repeated. This is also observed by  Devlin et al. (2015) on different test splits, but using retrieval-based and pipeline based methods for IC.

Intrigued by the results that almost all the representations end up with similar proportion of unique captions, we further investigate the models using a $k$-nearest neighbor approach. The key idea is that if the IC systems perform some form of image matching and a complex text retrieval from the training set, then the nearest neighbour (from training) of a test image should have a similar caption to the one generated by the model. We note that the model is clearly not performing text retrieval as the LSTM does generate novel captions, possibly by aggregating or 'averaging' the captions of similar images and performing some factorization.

| Model | Unique (%) |
|---|---|
| Bag of Objects | 29.5 |
| Top-k Class | 29.0 |
| Softmax | 28.7 |
| ResNet Pool5 | 28.8 |
| Human | 99.4 |

Table 4: Unique captions with beam = 1.

| Type | B | M | C | S |
|---|---|---|---|---|
| Freq. | 0.868 | 0.591 | 6.956 | 0.737 |
| Proj. | 0.912 | 0.634 | 7.651 | 0.799 |
| Exact (2301) | 1.000 | 1.000 | 10.000 | 1.000 |
| Freq. ($\neg$ Exact) | 0.757 | 0.498 | 4.337 | 0.512 |
| Proj. ($\neg$ Exact) | 0.837 | 0.560 | 5.638 | 0.628 |

Table 5: $k$-nearest neighbor experiment

To perform this experiment, we begin by generating captions for every training image using the bag of objects model (with frequency counts). We then compute the $k$-nearest training images for each given test image using both the bag of objects representation and its projection (Eq. 2). Finally, we compute the similarity score between the generated caption of the test image against all $k$-nearest captions. The similarity score measures how well a generated caption matches its nearest neighbour's captions. We expect the score to be high if the IC system generates an image similar to something 'summarized' from the training set.

We report our results in Table 5. We observe that overall the captions seem to closely match the captions of 5 nearest training images. Further analysis showed that 2301 captions had nearest images at a zero distance, i.e., the same exact representation was seen during at least 5 times in training (note that CIDEr gives a score of 10 only if the test caption and *all* references are the same). We found that among the non-exact image matches, the projected image representation better captures candidates in the training set than the bag of objects. We further analyze the captions and provide details in the appendix.

## 5 CONCLUSION

We hypothesized that IC systems essentially exploit a *distributional similarity* space to 'generate' image captions, by attempting to match a test image to similar training image(s) and generate an image caption from these similar images. Our study focused on the *image* side of image captioning: We varied the image representations while keeping the text generation component of an end-to-end CNN-RNN model constant. We found that regardless of the image representation, end-to-end IC systems seem to match images and generate captions in a visual-semantic subspace for IC. We conclude that:

- A sparse, low-dimensional **bags-of-objects representation** can be used as a tool to investigate the contribution of images in IC; we demonstrated that such a vector is sufficient for generating good image captions;
- End-to-end IC models are remarkably capable of separating structure from noisy input representations, as demonstrated by **pseudo-random vectors**;
- End-to-end IC models suffer virtually no significant loss in performance when a high dimensional representation is **factorized** to a lower dimensional space;
- End-to-end IC models have **learned a joint visual-textual semantic subspace** by clustering images with similar visual and linguistic information together;
- End-to-end IC models rely on test sets with a **similar distribution** as the training set for generating good captions;
- End-to-end IC models **repeatedly generate the same captions** by matching images in the joint visual-textual space and 'retrieving' a caption in the learned joint space.

All the observations above strengthen our distributional similarity hypothesis – that end-to-end IC performs image matching and generates captions for a test image from similar image(s) from the training set – rather than performing actual image understanding. Our findings provide novel insights into what end-to-end IC systems are actually doing, which previous work only suggests or hints at without concretely demonstrating the distributional similarity hypothesis. We believe our findings are important for the IC community to further advance image captioning in a more informed manner.

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

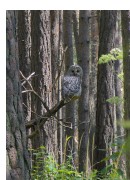

| Representation | CIDEr (△) | Caption |
|---|---|---|
| Bag of objects | 2.78 (+0.00) | a bird is perched on a branch in the sun . |
| VGG19 softmax | 3.14 (+0.36) | a owl is perched on a branch of a tree . |
| ResNet softmax | 3.67 (+0.89) | a owl is perched on a branch in a tree . |
| Places365 softmax | 2.00 (-0.77) | a bear is sitting on a branch in the wilderness . |
| Hybrid1365 softmax | 0.01 (-2.77) | a giraffe standing in a field of grass . |
| VGG19 fc7 | 0.18 (-2.59) | a black and white image of a bird sitting on a window sill . |
| ResNet pool5 | 0.38 (-2.40) | a large black bear standing in a forest . |
| Places365 pool5 | 0.34 (-2.43) | a giraffe standing in the middle of a forest . |
| Hybrid1365 pool5 | 3.03 (+0.26) | a bird is perched on a branch in a tree . |
| Embeddings | 2.38 (-0.40) | a bird sitting on a branch in a window . |

(a) Bag of objects: bird (1)

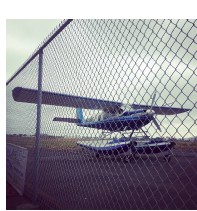

| Representation | CIDEr (△) | Caption |
|---|---|---|
| Bag of objects | 0.09 (+0.00) | a large airplane flying through a blue sky . |
| VGG19 softmax | 0.00 (-0.09) | a man in a baseball cap and sunglasses is holding a baseball bat . |
| ResNet softmax | 0.00 (-0.09) | a man is holding a baseball bat in a batting cage . |
| Places365 softmax | 0.06 (-0.03) | a dog is standing in the grass with a ball in its mouth . |
| Hybrid1365 softmax | 0.00 (-0.09) | a man holding a tennis racquet on a tennis court . |
| VGG19 fc7 | 0.73 (+0.63) | a plane is sitting on a runway with a few people . |
| ResNet pool5 | 0.01 (-0.08) | a train is on the tracks in a city . |
| Places365 pool5 | 0.00 (-0.09) | a giraffe standing in a fenced in enclosure . |
| Hybrid1365 pool5 | 0.01 (-0.08) | a man holding a baseball bat standing next to home plate . |
| Embeddings | 0.01 (-0.09) | a baseball player holding a bat on a field . |

(b) Bag of objects: airplane (1)

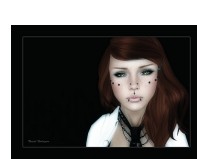

| Representation | CIDEr (△) | Caption |
|---|---|---|
| Bag of objects | 0.01 (+0.00) | a man wearing a suit and tie standing in front of a building . |
| VGG19 softmax | 0.04 (+0.04) | a woman in a pink wig and a pink dress . |
| ResNet softmax | 0.00 (-0.00) | a man in a suit and tie is smiling . |
| Places365 softmax | 0.13 (+0.12) | a woman with a red polka dotted dress tie . |
| Hybrid1365 softmax | 0.06 (+0.05) | a woman in a red dress is talking on a cell phone . |
| VGG19 fc7 | 0.24 (+0.24) | a woman with a cell phone in her hand . |
| ResNet pool5 | 0.08 (+0.08) | a woman in a red shirt and tie . |
| Places365 pool5 | 0.10 (+0.09) | a woman is holding a cell phone to her ear . |
| Hybrid1365 pool5 | 0.05 (+0.04) | a woman in a dress shirt and tie holding a parasol . |
| Embeddings | 0.00 (-0.01) | a man wearing a tie and a shirt and a tie . |

(c) Bag of objects: person (1), tie (1)

Figure 5: Example outputs from our system with different representations, the sub-captions indicate the annotation along with the frequency in braces. We also show the CIDEr score and the difference in CIDEr score relative to the *Bag of Objects* representation.

## A    ANALYSIS ON GENERATED CAPTIONS

Here, we provide a qualitative analysis of different image representations presented and gain some insights into how they contribute to the the IC task. The *Bag of Objects* representation led to a strong performance in IC despite being extremely sparse and low-dimensional (80 dimensions). Analyzing the test split, we found that each vector consists of only 2.86 non-zero entries on average (standard deviation 1.8, median 2). Thus, with the minimal information being provided to the generator RNN, we find it surprising that it is able to perform so well.

We compare the output of the remaining models against the *Bag of Objects* representation by investigating what each representation adds to or subtracts from this simple, yet strong model. We start by selecting images (from the test split) annotated with the exact same *Bag of Objects* representation – which should result in the same caption. For our qualitative analysis, several sets of one to three MSCOCO categories were manually chosen. For each set, images were selected such that there is exactly one instance of each category in the set and zero for others. We then shortlisted images where the captions generated by the *Bag of Objects* model produced the five highest and five lowest CIDEr scores (ten images per set). We then compare the captions sampled for each of the other representations.

Figure 5 shows some example outputs from this analysis. In Figure 5a, *Bag of Objects* achieved a high CIDEr score despite only being given "bird" as input, mainly by 'guessing' that the bird will

be perching/sitting on a branch. The object-based *Softmax* (VGG and ResNet) models gave an even more accurate description as "owl" is the top-1 prediction of both representations (96% confidence for VGG, 77% for ResNet). *Places365* predicted "swamp" and "forest". The *Penultimate* features on the other hand struggled with representing the images correctly. In Figure 5b, *Bag of Objects* struggled with lack of information (only "airplane" is given), the *Softmax* features mainly predicted "chainlink fence", Places365 predicted "kennel" (hence the dog description), and it most likely that *Penultimate* has captured the fence-like features in the image rather than the plane. In Figure 5c, the *Softmax* features generally managed to generate a caption describing a woman despite not explicitly containing the 'woman' category. This is because other correlated categories were predicted, such as "mask", "wig", "perfume", "hairspray" and in the case of *Places365* "beauty salon" and "dressing room".

## B   HYPERPARAMETER SETTINGS

Our model settings were:

- LSTM with 128 dimensional word embeddings and 256 dimensional hidden representations Dropout over LSTM of 0.8
- We used Adam for optimization.
- We fixed the learning rate to 4e-4

We report our results by keeping the above settings constant.

