# OpenReview forum: "What is image captioning made of?"
_ICLR.cc/2018/Conference — Reject_

### Official Review · AnonReviewer2 · 2017-11-22
**Needs work**

**Rating:** 4
**Confidence:** 5

**Review:**

This paper analyzes the effect of image features on image captioning. The authors propose to use a model similar to that of Vinyals et al., 2015 and change the image features it is conditioned on. The MSCOCO captioning and Flickr30K datasets are used for evaluation.

Introduction
- The introduction to the paper could be made clearer - the authors talk about the language of captioning datasets being repetitive, but that fact is neither used or discussed later.
- The introduction also states that the authors will propose ways to improve image captioning. This is never discussed.

Captioning Model and Table 1
- The authors use greedy (argmax) decoding which is known to result in repetitive captions. In fact, Vinyals et al. note this very point in their paper. I understand this design choice was made to focus more on the image side, rather than the decoding (language) side, but I find it to be very limiting. In this regime of greedy decoding it is hard to see any difference between the different ConvNet features used for captioning - Table 1 shows meteor scores within 0.19 - 0.22 for all methods.
- Another effect (possibly due to greedy decoding + choice of model), is that the numbers in Table 1 are rather low compared to the COCO leaderboard. The top 50 entries have METEOR scores >= 0.25, while the maximum METEOR score reported by the authors is 0.22. Similar trend holds for other metrics like BLEU-4.
- The results of Table 5 need to be presented and interpreted in the light of this caveat of greedy decoding.

Experimental Setup and Training Details
- How was the model optimized? No training details are provided. Did you use dropout? Were hyperparamters fixed for training across different feature sizes of VGG19 and ResNet-152? What is the variance in the numbers for Table 1?

Main claim of the paper
Devlin et al., 2015 show a simple nearest neighbor baseline which in my opinion shows this more convincingly. Two more papers from the same group which use also make similar observations - tweaking the image representation makes image captioning better: (1) Fang et al., 2015: Multiple-instance Learning using bag-of-objects helps captioning (2) Misra et al. 2016 (not cited): label noise can be modeled which helps captioning. This claim has been both made and empirically demonstrated earlier.

Metrics for evaluation
- Anderson et al., 2016 (not cited) proposed the SPICE metric and also showed how current metrics including CiDER may not be suitable for evaluating image captions. The COCO leaderboard also uses this metric as one of its evaluation metrics. If the authors are evaluating on the test set and reporting numbers, then it is odd that they `skipped' reporting SPICE numbers.

Choice of Datasets
- If we are thoroughly evaluating the effect of image features, doing so on other datasets is very important. Visual Genome (Krishnan et al., not cited) and SIND (Huang et al., not cited) are two datasets which are both larger than Flickr30k and have different image distributions from MSCOCO. These datasets should show whether using more general features (YOLO-9k) helps.
The authors should evaluate on these datasets to make their findings stronger and more valuable.

Minor comments
- Figure 1 is hard to read on paper. Please improve it.
- Figure 2 is hard to read even on screen. It is really interesting, so improving the quality of this figure will really help.

---

> ### Author Response · Authors · 2017-12-22
> **on specific comments**
>
> > - The introduction to the paper could be made clearer
>
> We have updated the introduction to make it clearer.
>
> > the authors talk about the language of captioning datasets being repetitive, but that fact is neither used or discussed later.
>
> In our analysis we observed that in all cases, i.e., using any type of representation, there is only a small subset (20-30%) of the captions that are unique. This was mentioned in section 4.5 of our original submission of the paper. We have further clarified this section in the updated version.
>
>
> > The introduction also states that the authors will propose ways to improve image captioning. This is never discussed.
>
> We do not promise to do that, but rather state that findings could help improve image captioning systems.
>
> >  Captioning Model and Table 1 - The authors use greedy (argmax) decoding which is known to result in repetitive captions. In fact, Vinyals et al. note this very point in their paper. I understand this design choice was made to focus more on the image side, rather than the decoding (language) side, but I find it to be very limiting.
> >  In this regime of greedy decoding it is hard to see any difference between the different ConvNet features used for captioning
>
> This was purposefully done for determinism. We wanted to understand the best 'choice of words' by the model given a particular representation.
>
>
> > The top 50 entries have METEOR scores >= 0.25, while the maximum METEOR score reported by the authors is 0.22.  Similar trend holds for other metrics like BLEU-4.
>
> Our model should be compared with the Neuraltalk model as it has the same settings. Other similar models (like Vinyals et al 2015) use ensembles and other engineering tricks that we are not interested in.
>
> > - The results of Table 5 need to be presented and interpreted in the light of this caveat of greedy decoding. Experimental Setup and Training Details - How was the model optimized? No training details are provided. Did you use dropout? Were hyperparameters fixed for training across different feature sizes of VGG19 and ResNet-152? What is the variance in the numbers for Table 1?
>
> Our settings are:
> LSTM with 128 dimensional word embeddings and 256 dimensional hidden representations
> Dropout over LSTM of 0.8
> Adam for optimization.
> Learning rate = 4e-4
> We’ll add the variance figures to an improved version of the paper.

---

> ### Author Response · Authors · 2017-12-22
> **on specific comments**
>
> > Main claim of the paper Devlin et al., 2015 show a simple nearest neighbor baseline which in my opinion shows this more convincingly. Two more papers from the same group which use also make similar observations - tweaking the image representation makes image captioning better: (1) Fang et al., 2015: Multiple-instance Learning using bag-of-objects helps captioning (2) Misra et al. 2016 (not cited): label noise can be modeled which helps captioning. This claim has been both made and empirically demonstrated earlier. Metrics for evaluation
>
> Once again, we use the object representations as a tool for our investigation. Our aim is not to improve on the task.
>
> >  - Anderson et al., 2016 (not cited) proposed the SPICE metric and also showed how current metrics including CiDER may not be suitable for evaluating image captions. The COCO leaderboard also uses this metric as one of its evaluation metrics. If the authors are evaluating on the test set and reporting numbel rs, then it is odd that they `skipped' reporting SPICE numbers.
>
> We have answered this before. We note however that our observations are also consistent with the numbers on the SPICE metric.
>
> >  Choice of Datasets - If we are thoroughly evaluating the effect of image features, doing so on other datasets is very important. Visual Genome (Krishnan et al., not cited) and SIND (Huang et al., not cited) are two datasets which are both larger than Flickr30k and have different image distributions from MSCOCO. These datasets should show whether using more general features (YOLO-9k) helps. The authors should evaluate on these datasets to make their findings stronger and more valuable.
>
> SIND represents a very different type of data where sentences compose a narrative. Different kinds of models are needed and these are evaluated using different metrics. Visual Genome, on the other hand, is a subset of MSCOCO with different kind of annotations (object specific captions). We are interested in investigating the CNN-LSTM model in this paper, and while it may be applied to a different domain of the same task (e.g. image captioning on Flickr30k), it is not clear how this can be applied directly to a different set of tasks.
>
>
> > Minor comments - Figure 1 is hard to read on paper. Please improve it. - Figure 2 is hard to read even on screen. It is really interesting, so improving the quality of this figure will really help.
>
> We have enlarged the Figure 1.
>
> We initially planned to add the full, high-resolution versions of Figure 2 in the appendix. Unfortunately each t-SNE visualisation was around 18MB -- which will increase the file size to over 100MB if we were to add all images (3 pairs before-after projection). We have added an anonymised external link in the updated version of the paper. The images can now be found here: https://github.com/anonymousiclr/HJNGGmZ0Z

---

### Official Review · AnonReviewer3 · 2017-12-01
**Not clear contributions, lack of comparisons with other methods and weak results.**

**Rating:** 4
**Confidence:** 4

**Review:**

The paper claims that image captioning systems work so well, while most recent state of the art papers show that they produce 50% errors, so far from perfect.

The paper lacks novelty, just reports some results without proper analysis or insights.

Main weakness of the paper:
 - Missing many IC systems citations and comparisons (see https://competitions.codalab.org/competitions/3221#results)
 - According to "SPICE: Semantic Propositional Image Caption Evaluation" current metrics used in image captioning don't correlate with human judgement.
- Most Image Caption papers which use a pre-trained CNN model, do fine-tune the image feature extractor to improve the results (see Vinyals et al. 2016). Therefore correlation of the image features with the captions is weaker that it could be.
- The experiments reported in Table1 are way below state-of-the-art results, there a tons of previous work with much better results, see https://competitions.codalab.org/competitions/3221#results
 - To provide a fair comparison authors, should compare their results with other paper results.
 - Tables 2 and 3 are missing the original baselines.
The evaluation used in the paper don't correlate well with human ratings see (SPICE paper), therefore trying to improve them marginally doesn't make a difference.
- Getting better performance by switching from VGG19 to ResNet152 is expected, however they obtain worse results than Vinyals et al. 2016 with inception_v3.
- The claim "The bag of objects model clusters these group the best" is not supported by any evidence or metric.

One interesting experiment but missing in section 4.4 would be how the image features change after fine-tuning for the captioning task.


Typos:
 - synsest-level -> synsets-level

---

> ### Author Response · Authors · 2017-12-22
> **on specific comments**
>
> > The paper lacks novelty, just reports some results without proper analysis or insights.
> > Main weakness of the paper:
> > - Missing many IC systems citations and comparisons (see https://competitions.codalab.org/competitions/3221#results)
>
> We stress that our evaluations are with respect to the model proposed by Karpathy et al 2015. Our goal is not to 'beat or break' systems but to understand the 'whys' and 'hows'.
>
> >  - According to "SPICE: Semantic Propositional Image Caption Evaluation" current metrics used in image captioning don't correlate with human judgement.
>
> We are not claiming explicitly that any of the metrics has good correlation with human judgements. As we mentioned before our focus on CIDEr is because a) the official evaluation script from MSCOCO  contains only CIDEr, Meteor, BLEU and ROUGE, b) CIDEr is a metric that was officially developed for the task of image captioning, c) CIDEr is the official metric for MSCOCO, d) papers by Liu et al 2017, Kilickaya et al. 2017 and Vedantam et al, 2015 (with the human correlation experiments over Flickr8k dataset) still state the importance of CIDEr as a metric for image captioning. We further note that we observe a similar trend as we found in CIDEr, so all our observations are still valid.
>
>
> >  - Most Image Caption papers which use a pre-trained CNN model, do fine-tune the image feature extractor to improve the results (see Vinyals et al. 2016). Therefore correlation of the image features with the captions is weaker that it could be.
>
> While it is true that fine-tuning could have been helpful to bump performance, our paper deals with an exploration of representational properties. Vinyals et al. 2016 has shown that fine-tuning gives only a minor 1-point improvement for BLEU. This is also using an ensemble of models. We again state that our experiments are about understanding image captioning models.
>
> > - To provide a fair comparison, authors should compare their results with other paper results. - Tables 2 and 3 are missing the original baselines.
>
> We will add the results from the comparable papers, even though our focus is not comparisons or to show performance improvements over other models. However, we do not understand what the reviewer means by “original baselines”. Could you please clarify?
>
> > The evaluation used in the paper don't correlate well with human ratings see (SPICE paper), therefore trying to improve them marginally doesn't make a difference.
>
> Please see answer above regarding metrics. In addition, our focus is not to improve the performance of the system, but to interpret the 'how' and 'why' of the system. To this end, we have made significant progress.
>
> >  - Getting better performance by switching from VGG19 to ResNet152 is expected, however they obtain worse results than Vinyals et al. 2016 with inception_v3.
>
> We have not chosen Vinyals et al. 2016 since it uses ensembles and other clever engineering tricks. This would make it hard to answer the questions we ask in this paper -- namely, the contribution of image representation. Our results are comparable to those in Karpathy et al, 2015. We will add this into the table.
>
> > - The claim "The bag of objects model clusters these group the best" is not supported by any evidence or metric.
>
> We believe that the reviewer has misunderstood the sentence. This sentence explains the observations in Figure 1 (more specifically Figure 1a). The figure shows that the bag of objects representation forms better clusters. It shows the cosine distances between each group for the bag of objects representation. We see from the figure that the bag of objects representations clusters these groups best. For example, the average image representation of “dog” correlates with images containing “dog” as a pair like “dog+person” and “dog+toilet”. We are aware that this is true for our given example, however we expect this to extrapolate over other examples in the dataset.
>
>
>
> > One interesting experiment but missing in section 4.4 would be how the image features change after fine-tuning for the captioning task.
>
> We will do it as a future work, even though this does not allow us to answer our questions posed in this paper.

---

### Official Review · AnonReviewer1 · 2017-12-02
**Clear rejection.**

**Rating:** 4
**Confidence:** 5

**Review:**

This paper is an experimental paper. It investigates what sort of image representations are good for image captioning systems.

Overall, the idea seems relevant and there are some good findings but I am sure that image captioning community is already aware of these findings.

The main issue of the paper is the lack of novelty. Even for an experimental paper, I would argue that novelty in the experimental methodology is an important fact. Unfortunately, I do not see any novel concept in the experimental setup.

I recomend this paper for a workshop presentation.

---

> ### Author Response · Authors · 2017-12-22
> **on comments**
>
> > Overall, the idea seems relevant and there are some good findings but I am sure that image captioning community is already aware of these findings.
> > The main issue of the paper is the lack of novelty. Even for an experimental paper, I would argue that novelty in the experimental methodology is an important fact.
>
> Our claim is in the novel 'insights' into end-to-end model of image captioning models. Our empirical evaluations with multiple representations, visualizations and out of domain experiments reveal new and important insights that should be of interest to the community.
>
> We kindly ask clarification from the reviewer regarding what is meant by 'novelty in experimental methodology'.

---

### Public Comment · ~Abhisek_Konar1 · 2017-12-16
**Report on reproducibility of the paper**

In this report, the ﬁndings of this paper submitted to the ICLR 2018 Conference were attempted to be replicated. In the process of replication, two major components were identiﬁed. The ﬁrst breakdown included building the baseline model. The second subsection contained the core of the research, which was to answer the key questions and address which image transformations affected the accuracy of neural image captioning systems. Following the steps outlined in the paper as closely as possible, we were able to build a very similar baseline model, and perform three of the ﬁve image transformations that were speciﬁed.
The base line model used in the paper is a combination of the approaches of Karpathy [1] and Vinyals [2]. We were able to closely replicate that model by breaking down it into 3 subsets, a combination of an image model and a language model with a CNN used as an encoder of the images, and an LSTM for the language model as mentioned in the paper.
For the image transformations, we were able to successfully reproduce three out of the ﬁve: penultimate layer extraction, class prediction vector, and object-class word embeddings. For the penultimate layer extraction, we implemented the pretrained VGG19 and ResNet152 models. The VGG19 uses very small convolutional ﬁlters and uses very deep weight layers of up to 19. The ResNet152 model, as implemented by He et al.[9], uses 8 times deeper nets than VGG19. Both the models were implemented via the Keras distribution with a TensorFlow backend.
The class prediction vector transformation involved investigating more complex image representations, where the vector elements are now estimated posterior probabilities of the possible object categories. To obtain these posterior distribution vectors, the pre-trained network ResNet152 was again used to retrieve a 1000 dimensional posterior vector.
The last transformation we were able to replicate was the object-class word embeddings. This procedure is carried out over the entire 1000 dimensional output of the Softmax layer of pre-trained model ResNet152 where all the procured word2vec representations are ﬁnally averaged. This averaged vector acts as the image representation for the image model.
The evaluation metric used for the score calculation nltk based corpus BLEU introduced by Papineni et al. [12]. Using a beam size of 1, as done by the authors, a steady rise was observed in the corpus BLEU score for all three representations. Penultimate layer and softmax implementations outperformed the word2vec image representation which had BLEU scores ranging between 0.7540 and 0.4646 from BLEU-2 to BLEU4. For both penultimate and softmax image representations, ResNet152 performed better than VGG19 with BLEU scores ranging from 0.5598 to 0.9216 for softmax and 0.5889 to 0.8937 for penultimate with BLEU varying from 4 to 1. It was only marginally better than VGG19’s BLEU 4-1 scores ranging between 0.5346 and 0.9158 for softmax and 0.5962 and 0.8524 for penultimate.
One caveat of this report was that it was not feasible to train the model on the MSCOCO dataset as the paper. This was due to computational restrictions, as training a model on the Flickr8K dataset, which is much smaller than the MSCOCO dataset, took a K80 equipped server approximately 2 days for a small batch size. Due to the inability to use the MSCOCO, we experienced two drawbacks during the replication; The ﬁrst included hindering our ability to implement the 4th and 5th image transformations, and the second was fact that we were not able to reproduce an exact copy of the works presented by the authors. Although we used a different dataset, we still noticed similar trends in the ones obtained by the tests carried out in the MSCOCO dataset. For example, both our tests and the original authors’ test both had the ResNet152 pre-trained network slightly outperforming the VGG19 network in the different image transformations.

---

### Author Response · Authors · 2017-12-22
**Rebuttal**

We thank the reviewers for the comments.

Our submission is based on the simple end-to-end model as proposed by Karpathy et al 2015. We use this model because its simplicity makes it easier to focus on the image component. We are interested in the interpretability of the image-captioning system rather than the performance on the task. In addition, more advanced models can be considered similar variants of Karpathy et al 2015. We do not claim novelty with respect to the captioning model. Instead, our submission presents novel insights into the image captioning task which we are confident that it should be of interest to the community. Also as our submission involves work on understanding the representational contributions, we consider our work highly relevant to the conference on learning representations. Our main contributions are:

1) We show that the image-conditioned language model implicitly learns and exploits a joint image representation and language semantic space instead of actually understanding images (sections 4.2, 4.3, 4.4).

2) Our experiments with factorized and compressed image embeddings (section 4.1) reveals that the models do not benefit from the full representational space. We observe that the performance of the model trained with a 2048 dimensional representation is nearly identical to the performance of the model trained with a compressed 80-dimensional representation virtually resulting in ‘no information loss’.

3) The experiments with pseudorandom representations (section 3.2) reveal that the end-to-end models learn to separate structure from noisy representations in the framework and exploit it to produce near ideal performance, i.e., the performance with structured representations versus the performance with noisy representations is similar.


The reviewers also raised concern regarding the absence of SPICE as a metric for evaluation. We focus on CIDEr because: a) the metrics in the official evaluation script from MSCOCO contains support for only CIDEr, Meteor, BLEU and ROUGE; b) CIDEr is a metric that was officially developed for the task of image captioning, and is supposed to be the official metric for MSCOCO; c) papers by Liu et al 2017, Kilickaya et al. 2017 and Vedantam et al, 2015 (with the human correlation experiments over Flickr8k dataset) still state the importance of CIDEr as a metric for image captioning. However, we will provide the results on SPICE in the revised version. We also note that a similar trend is observed with SPICE.

* Liu et al. (ICCV 2017) Improved Image Captioning via Policy Gradient Optimization of SPIDEr
* Kilickaya et al. (EACL 2017) Re-evaluating Automatic Metrics for Image Captioning
* Vedantam et al. (CVPR 2015) CIDEr: Consensus-based Image Description Evaluation

---

### Author Response · Authors · 2017-12-22
**Updates**

We have updated the paper with these salient changes:

* re-written introduction
* updated results with SPICE
* updated sections 4.3, 4.4 and 4.5 with more support to claims
* re-written conclusion

---

### Comment · AnonReviewer2 · 2018-01-23
**Final Review - Rejection**

After reading the other reviews, the discussion and the revised paper, I am not convinced of the contributions of the paper (even if I were to ignore the weakness in the experimental setup, as I explain later).
Let's focus on the conclusion section of the paper (page 11) to see what the authors claim.

- "End-to-end IC models are remarkably capable of separating structure from noisy input representations, as demonstrated by pseudo-random vectors": This statement is factually incorrect and its empirical implications are not surprising. The pseduo random vectors (pg 5) are not noisy at all! First, they are deterministic mapping from image space to a vector space (this is the definition of a feature extractor). Secondly, and more importantly, they are generated using Gold (or ground-truth) object counts. By the very definition of this, they are NOT noisy! Your conclusion that such a representation works well for image captioning is not surprising. To any system using image features, all it really cares about is how good the mapping from image to vector (feature) space is. Your mapping is defined using ground truth counts. This claim is repeated throughout the paper - abstract, introduction and conclusion. In all three repititions, the authors claim incorrectly that the representation is `noisy'.
- "A sparse, low-dimensional bags-of-objects representation can be used as a tool to investigate the contribution of images in IC; we demonstrated that such a vector is sufficient for generating good image captions": Like I had mentioned in my review earlier - bag-of-objects based representations have been shown to be sufficient for generating good image captions (Fang et al., 2015).
- "End-to-end IC models repeatedly generate the same captions by matching images in the joint visual-textual space and ‘retrieving’ a caption in the learned joint space": Again, a similar claim was empirically shown by Devlin et al., by using a nearest neighbor technique for image captioning. They also showed that such a simple technique outperformed end-to-end captioning systems.
- "End-to-end IC models rely on test sets with a similar distribution as the training set for generating good captions": Please look at Sec 4.3.3 of "Show and Tell: Lessons Learned from the 2015 MSCOCO Image Captioning Challenge" where transfer experiments are shown across datasets, showing that similarity between train and test distributions is important from transfer.

There are also a few issues with the experimental setup.
- Table 1 shows nearly constant numbers across B-4, M, and S.
- Table 1 shows that using Pool5 features for ResNet-152 performs better than softmax for ResNet-152. Now Figure 1 shows us that softmax ResNet-152 is better at discriminating between images based on object groups rather than Pool5 ResNet-152. A similar negative correlation exists between Table 1 and Figure 1 for the pairs (softmax ResNet-152, pseudo-random). The reason for introducing Figure 1 as stated in the paper is "If the representation is informative for IC, then the representations should ideally semantically related images together, and in turn allow for relevant captions to be generated." This statement is clearly falsified by the pairs (and many more such exist within your results).

I do not think this paper is ready for publication and stick with "needs work".

---

> ### Author Response · Authors · 2018-01-25
> **On final review (part 2)**
>
> > - "End-to-end IC models repeatedly generate the same captions by matching images in the joint visual-textual space and ‘retrieving’ a caption in the learned joint space": Again, a similar claim was empirically shown by Devlin et al., by using a nearest neighbor technique for image captioning. They also showed that such a simple technique outperformed end-to-end captioning systems.
>
> Devlin et al. only show that a `nearest neighbour method’ works as well as end-to-end captioning systems. We show that end-to-end IC *is* sort of a nearest neighbour retrieval in the joint space. These are two completely different conclusions. We again respectfully disagree with the reviewer that the two are same or similar.
>
> > - "End-to-end IC models rely on test sets with a similar distribution as the training set for generating good captions": Please look at Sec 4.3.3 of "Show and Tell: Lessons Learned from the 2015 MSCOCO Image Captioning Challenge" where transfer experiments are shown across datasets, showing that similarity between train and test distributions is important from transfer.
>
> We are aware of this and have explicitly mentioned Vinyals et al 2016 in Sec 4.4 of the our paper. We have looked carefully again at Vinyals et al 2016 - they only mention that the ``BLEU scores degrade by 10 points’’ and merely suggest that this *could* be because ``more differences in vocabulary and a larger mismatch’’. In our paper, we demonstrated that what they suggested is not exactly true. We show that there are only 8.6% out of vocabulary words in Flickr30k (around 8% vocabulary mismatch is true in the MSCOCO train v/s dev set). So there is more to the performance drop than just vocabulary mismatch. And thus, we further show that it is also because the `object’ distribution (from the image side) in MSCOCO is almost identical in train and test, compared to the larger differences between MSCOCO train vs. Flickr30k test. That is, the `types’ of images are maintained in MSCOCO evaluation sets, while they vary in Flickr30k test set. To our knowledge this is still a novel and an important claim.
>
>
> > There are also a few issues with the experimental setup.
> > - Table 1 shows nearly constant numbers across B-4, M, and S.
>
> We do not understand why constant metric scores can indicate that it’s an “issue with our experimental setup”. In fact, we think the constant scores further strengthen our claim that the metrics do not capture what exactly happens as shown by several other papers that we cite.
>
>
> > - Table 1 shows that using Pool5 features for ResNet-152 performs better than softmax for ResNet-152. Now Figure 1 shows us that softmax ResNet-152 is better at discriminating between images based on object groups rather than Pool5 ResNet-152. A similar negative correlation exists between Table 1 and Figure 1 for the pairs (softmax ResNet-152, pseudo-random). The reason for introducing Figure 1 as stated in the paper is "If the representation is informative for IC, then the representations should ideally semantically related images together, and in turn allow for relevant captions to be generated." This statement is clearly falsified by the pairs (and many more such exist within your results).
>
> We state ‘If the representation is informative for IC, then the representations should ideally semantically related images together, and in turn allow for relevant captions to be generated’ as our hypothesis and we follow it up in the next section. Figure 1 merely shows how well the initial representations cluster with cosine distance as the metric. We don’t make any conclusions using Figure 1.
>
> As our work mainly the deals with the evaluation of representational contributions we consider ICLR the best venue to disseminate our findings.

---

> ### Author Response · Authors · 2018-01-25
> **On Final Review (part 1)**
>
> We appreciate and thank the reviewer for going through our rebuttal and revised manuscript and for the comments. However, we disagree with many of the points raised in the above review.
>
> > - "End-to-end IC models are remarkably capable of separating structure from noisy input representations, as demonstrated by pseudo-random vectors": This statement is factually incorrect and its empirical implications are not surprising. The pseduo random vectors (pg 5) are not noisy at all! First, they are deterministic mapping from image space to a vector space (this is the definition of a feature extractor). Secondly, and more importantly, they are generated using Gold (or ground-truth) object counts. By the very definition of this, they are NOT noisy! Your conclusion that such a representation works well for image captioning is not surprising. To any system using image features, all it really cares about is how good the mapping from image to vector (feature) space is. Your mapping is defined using ground truth counts. This claim is repeated throughout the paper - abstract, introduction and conclusion. In all three repititions, the authors claim incorrectly that the representation is `noisy'.
>
>
> We claim pseudo-random vectors as being noisy representations as they are not `just’ one to one mapping from the image to vector feature space but they are actually a composition of object vectors (where objects are represented by random vectors). The composition specifically involves addition and the information about the number of occurrences of the objects --specifically: multiplication of random vectors per object by the number of object occurrences and then addition of vectors across multiple objects. The resultant composition is `noisy’ -- this can be seen both in Figure 1, where the initial representations are shown to not form any clusters, as well as in Figure 2(d), where the initial representations again form no clusters. We kindly refer the reviewer to the full tSNE plot for the initial representations of pseudo-random vectors here: https://github.com/anonymousiclr/HJNGGmZ0Z/blob/master/tsne_initial_pseudorandom_4000.png. Further, the conclusion doesn’t change even if we use predicted counts.
>
> Despite the review strongly stating it to be factually incorrect, we stand by our conclusions. We have repeated our claims of the representations in the projected space. We observe that the resultant framework has, to some extent, captured the compositional operation, despite the initial representation being difficult to decipher.
>
>
> > - "A sparse, low-dimensional bags-of-objects representation can be used as a tool to investigate the contribution of images in IC; we demonstrated that such a vector is sufficient for generating good image captions": Like I had mentioned in my review earlier - bag-of-objects based representations have been shown to be sufficient for generating good image captions (Fang et al., 2015).
>
> Firstly, Fang et al 2015 does not use neural end-to-end image captioning. Secondly, they used a high 1000-dimensional "bag of surface-level-text-labels taken directly from captions" space, while we show a *low* barely 80-dimensional *object category* space (that is objects occuring in the images) performs as well. We disagree with the reviewer that Fang et al have the same conclusions as ours.

---

### Decision · Program_Chairs · 2018-01-29
**ICLR 2018 Conference Acceptance Decision**

**Decision:**

Reject

**Comment:**

Paper reviewed by three experts who have provided detailed feedback. All three recommend rejection, and this AC sees no reason to overrule their recommendation.